# Efficacy and Effect on Liver Functional Reserve of Atezolizumab and Bevacizumab for Unresectable Hepatocellular Carcinoma in Patients Who Do Not Meet Eligibility Criteria of IMbrave150

**DOI:** 10.3390/cancers14163938

**Published:** 2022-08-15

**Authors:** Takuya Sho, Goki Suda, Yoshiya Yamamoto, Ken Furuya, Masaru Baba, Koji Ogawa, Akinori Kubo, Yoshimasa Tokuchi, Qingjie Fu, Zijian Yang, Megumi Kimura, Takashi Kitagataya, Osamu Maehara, Shunsuke Ohnishi, Akihisa Nakamura, Ren Yamada, Masatsugu Ohara, Naoki Kawagishi, Mitsuteru Natsuizaka, Masato Nakai, Kazuharu Suzuki, Takaaki Izumi, Takashi Meguro, Katsumi Terashita, Tomofumi Takagi, Jun Ito, Tomoe Kobayashi, Takuto Miyagishima, Naoya Sakamoto

**Affiliations:** 1Departments of Gastroenterology and Hepatology, Graduate School of Medicine, Hokkaido University, Sapporo 060-0808, Japan; 2Hakodate City Hospital, Hokkaido 041-8680, Japan; 3Department of Gastroenterology and Hepatology, Japan Community Health Care Organization (JCHO) Hokkaido Hospital, Hokkaido 062-8618, Japan; 4Laboratory of Molecular and Cellular Medicine, Faculty of Pharmaceutical Sciences, Hokkaido University, Sapporo 060-0808, Japan; 5Sapporo City General Hospital, Hokkaido 060-8604, Japan; 6Hokkaido Gastroenterology Hospital, Hokkaido 065-0041, Japan; 7Japan Community Health Care Organization (JCHO) Sapporo Hokushin Hospital, Hokkaido 004-8618, Japan; 8The Hokkaido Medical Center, Hokkaido 063-0005, Japan; 9Tomakomai City Hospital, Tomakomai 053-8567, Japan; 10Kushiro Rosai Hospital, Hokkaido 085-8533, Japan

**Keywords:** atezolizumab, bevacizumab, eligibility criteria of IMbrave150, hepatocellular carcinoma

## Abstract

**Simple Summary:**

The IMbrave150 trial led to the approval of atezolizumab and bevacizumab for the treatment of unresectable hepatocellular carcinoma (HCC). We performed a retrospective multicenter study including 115 patients with unresectable HCC treated with atezolizumab and bevacizumab, revealing that the combination of atezolizumab and bevacizumab is equally effective for patients meeting the IMbrave150 trial eligibility criteria and for patients not meeting these criteria, generally due to a history of systemic therapy, platelet counts < 75 × 10^9^/L, Child-Pugh B, and 2+ proteinuria. However, liver functional reserve should be carefully monitored in patients not meeting the IMbrave150 trial eligibility criteria.

**Abstract:**

The IMbrave150 trial demonstrated the high efficacy and safety of atezolizumab and bevacizumab for unresectable hepatocellular carcinoma (HCC). In this multicenter study, the efficacy of this combination and its effect on liver functional reserve were evaluated in patients not meeting the eligibility criteria of IMbrave150. Of 115 patients with unresectable HCC treated with atezolizumab and bevacizumab between October 2020 and January 2022, 72 did not meet the eligibility criteria of IMbrave150, most frequently due to a history of systemic therapy (60/72), platelet counts < 75 × 10^9^/L (7/72), Child-Pugh B (9/72), and 2+ proteinuria (8/72). Atezolizumab and bevacizumab therapy was equally effective for patients who did or did not meet the eligibility criteria (PFS, 6.5 vs. 6.9 months, *p* = 0.765), consistent with subgroup analyses of histories of systemic therapy, platelet counts, Child-Pugh, and proteinuria. Baseline ALBI scores were worse in patients who did not meet the criteria than in those who did and significantly worsened after treatment initiation in patients not meeting the criteria (baseline vs. 12 weeks; 2.35 ± 0.43 vs. −2.18 ± 0.54; *p* = 0.007). Accordingly, atezolizumab plus bevacizumab was effective for patients not meeting the eligibility criteria of IMbrave150, although careful monitoring for changes in liver functional reserve is needed.

## 1. Introduction

Recent advances in systemic therapy have dramatically changed the treatment landscape for unresectable hepatocellular carcinoma (HCC), with prolonged overall survival (OS) [1,2,3,4]. Various systemic therapies have been developed for unresectable HCC, including the multikinase inhibitors sorafenib [1], regorafenib [5], cabozantinib [6], and lenvatinib [2], an antibody against VEGFR2, ramucirumab [7], and the combination of the programmed death ligand 1 (PD-L1) inhibitor atezolizumab and the VEGF inhibitor bevacizumab. Atezolizumab and bevacizumab combination therapy for patients with unresectable HCC significantly prolonged OS over that of patients treated with sorafenib in the phase 3 clinical trial IMbrave150 [3]. Therefore, recent guidelines recommend the combination of atezolizumab and bevacizumab as a first-line systemic therapy for patients with unresectable HCC [8,9].

However, in real-world settings, many patients do not meet eligibility criteria of IMbrave150 due to a history of systematic therapy for unresectable HCC, protein uremia, anemia, or low platelet counts [3]. Thus, the accumulation of real-world data for the efficacy of atezolizumab and bevacizumab in unresectable HCC, especially in patients who do not meet eligibility criteria of IMbrave150, is required. We have recently reported the high efficacy and safety of atezolizumab and bevacizumab for unresectable HCC in patients in the early phase of treatment who do not meet the eligibility criteria of IMbrave150 [10]. However, Tiago de Castro et al. have recently reported that OS and progression-free survival (PFS) were significantly longer in patients who met the eligibility criteria of IMbrave150 than in patients who did not meet the eligibility criteria of IMbrave150 [OS: 15.0 months vs. 6.0 months; PFS: 8.7 months vs. 3.7 months, respectively] [11]. Thus, additional data for the efficacy of atezolizumab and bevacizumab in patients who do not meet IMbrave150 eligibility criteria are urgently needed.

In the treatment of HCC, liver functional reserve is one of the most important factors for treatment decision-making. Given the availability of various therapeutic options for unresectable HCC, the maintenance of hepatic functional reserve during systemic therapy is crucial for subsequent salvage therapy after progressive disease. Terashima et al. have reported that in patients with unresectable HCC treated with tyrosine kinase inhibitor (TKI), a longer OS is highly associated with post-progression survival, not PFS [12]. Thus, determining the factors associated with changes in liver functional reserve during atezolizumab and bevacizumab combination therapy is a clinically important issue. 

In this real-world multicenter study, we compared the efficacy of bevacizumab and atezolizumab and its effect on liver functional reserve in patients who did and did not meet the eligibility criteria for IMbrave150.

## 2. Methods

### 2.1. Patients and Study Design

This was a retrospective multicenter study. Consecutive unresectable HCC patients who were treated with atezolizumab and bevacizumab were recruited between October 2020 and February 2022 at the institutes participating in the NORTE Study Group [13,14,15,16,17]. Patients were included if they were treated with atezolizumab and bevacizumab between October 2020 and February 2022 and if sufficient clinical information was available. Clinical data were collected, including age, gender, blood tests, tumor markers, Child–Pugh score, albumin-bilirubin (ALBI) grade [18], modified ALBI (mALBI) grade [19], etiology of HCC, and Barcelona Clinic Liver Cancer (BCLC) stage. Patients were excluded if they had insufficient clinical data, had decompensated liver cirrhosis, or declined to participate in this study. All included patients were evaluated, using endoscopy, for the presence of varices before initiation of atezolizumab and bevacizumab, and, when necessary, the varices were properly treated.

Each attending physician typically evaluated the patients every 3 weeks by laboratory data and physical findings, and evaluated the treatment response every 6 to 12 weeks by dynamic CT or MRI according to Response Evaluation Criteria in Solid Tumors 1.1 (RECIST 1.1) and modified Response Evaluation Criteria in Solid Tumors (mRECIST) [20]. Adverse events (AEs) were evaluated by the attending physician every 3 weeks. Atezolizumab and/or bevacizumab was interrupted if grade 3 or higher AEs or unacceptable AEs were observed until AEs resolved. Atezolizumab and/or bevacizumab were resumed according to the package inserts. Atezolizumab and bevacizumab was discontinued when progressive disease (PD) was observed or when unacceptable AEs were observed.

AE grades were defined referring to the American Society of Clinical Oncology Clinical Practice Guidelines [21] and the National Cancer Institute Common Terminology Criteria for Adverse Events (CTCAE; version 4.0).

PFS, treatment responses, and changes in liver functional reserve were evaluated in patients with unresectable HCC treated with atezolizumab and bevacizumab stratified according to IMbrave150 eligibility and clinical factors.

The study conformed to the ethical guidelines of the Declaration of Helsinki and all participating patients provided informed consent. This study was approved by the ethics committee of Hokkaido University Hospital (020-0267) and by the ethical committee of each participating institution. 

### 2.2. Treatment Protocol

Every 3 weeks, patients were treated with 1200 mg of atezolizumab (Chugai Co. Ltd., Tokyo, Japan) and 15 mg/kg bevacizumab (Chugai Co. Ltd., Tokyo, Japan).

### 2.3. Evaluation of Liver Functional Reserve during Treatment

Changes in liver functional reserve were evaluated based on the ALBI score at baseline and 3, 6, 9, and 12 weeks after treatment initiation. The changes in mALBI grade between baseline and 12 weeks after treatment initiation were also evaluated. Referring to previous studies, the modified ALBI (mALBI) grade was evaluated by dividing ALBI grade 2 into 2a and 2b, using an ALBI score cut-off value of −2.270 [19].

### 2.4. Statistical Analysis

We analyzed continuous variables by Mann–Whitney U-test or the paired *t*-test. We analyzed categorical data using the Fisher’s exact test or chi-squared test. Survival curves for PFS were calculated by the Kaplan–Meier analysis and compared using the log-rank test. In this study, we did not analyze the overall survival due to insufficient follow-up duration.

In this study, we set *p* < 0.05 as statistically significant. We utilized SPSS Statistics 22.0 (IBM Corp., Armonk, NY, USA) in all analyses.

## 3. Results

### 3.1. Overview of Patient Characteristics According to Eligibility Criteria of IMbrave150

Between October 2020 and January 2022, 115 patients with unresectable HCC treated with atezolizumab and bevacizumab at the institutes of the NORTE study group were included in this study. Baseline patient characteristics are shown in Table 1. The median age was 72 years (range, 31–89 years). The majority of patients were male (95 males (82.6%) and 20 females (17.4%)). A total of 80 (69.6%) patients had BCLC stage C, and HBV, HCV and non-B non-C were identified in 35 (30.4%), 21 (18.3%), and 59 patients (51.3%), respectively. A total of 77 patients (67.0%) had ALBI grade 2, and 106 (92.2%) and 9 (7.8%) had a baseline Child-Pugh grade of A and B, respectively. All patients with Child-Pugh grade B at the initiation of treatment had Child-Pugh grade A at the decision-making point for the treatment of unresectable HCC.

Of 115 patients, 72 (62.6%) did not meet the eligibility criteria of IMbrave150. Patients did not meet the IMbrave150 eligibility criteria due to a history of TKI treatment (83.3% 60/72), platelet counts < 75 × 10^9^/L (9.7% 7/72); AST or ALT values exceeding 5 times the upper limit of normal (ULN), (5.6% 4/72 and 2.8% 2/72), Child-Pugh B (12.5% 9/72); serum creatinine > 1.5 times the ULN (4.2% 3/72), 2+ or 2+ < proteinuria (11.1% 8/72), and neutrophil count < 1500/mm^3^ (8.3% 6/72). A comparison of baseline characteristics between patients who did or did not meet the IMbrave150 eligibility criteria is shown in Table 1. 

### 3.2. Progression-Free Survival and Associated Factors in Patients Who Did or Did Not Meet the IMbrave150 Eligibility Criteria

As shown in Figure 1, the median PFS was 6.6 months (95% confidence interval (CI) 5.2–8.9 months). As shown in Figure 2, the median PFS was similar in patients who did and did not meet the eligibility criteria of IMbrave150 (median PFS 6.5 months (95% CI; 3.7–NE) vs. 6.9 months (95% CI; 4.2–8.9), HR 1.085 (95% CI 0.633–1.862); *p* = 0.795). Among the main factors preventing IMbrave150 eligibility, the median PFS were similar in patients with and without a history of systemic therapy (median PFS 6.6 months (95% CI; 3.7–NA months) vs. 6.9 months (95% CI; 4.2–8.9 months), HR 1.082 (95% CI 0.643–1.820); *p* = 0.766), patients with or without Child-Pugh B (6.0 months (95% CI, 1.6–NA months) vs. 6.6 months (95% CI, 4.2–8.8 months), HR 0.998 (95% CI, 0.399–2.492); *p* = 0.996), patients with or without 2+ proteinuria or more than 2+ proteinuria (NA (95% CI, 1.8–NA) vs. 6.9 months (95% CI, 5.2–8.9 months), HR 0.722 (95% CI, 0.173–3.008); *p* = 0.241), and patients with or without platelet < 7.5 × 10^4^/μL (median PFS NA (95% CI, 1.9–NA months) vs. 6.6 months (95% CI, 5.2–8.8 months), HR 1.658 (95% CI, 0.507–5.422); *p* = 0.921).

Subsequently, we conducted subgroup analyses. As shown in Figure 2, median PFS was similar between patients with an etiology of viral hepatitis and non-viral hepatitis, BCLC B and C, and mALBI grade 1–2a and 2b.

Patients treated with atezolizumab and bevacizumab as a third-line or further lines therapy had a significantly shorter PFS than that of patients treated with atezolizumab and bevacizumab as a first- or second-line therapy (4.2 months (95% CI 2.8–6.9 months) vs. 7.6 months (95% CI 5.4–10.3 months), HR 1.884 (95% CI 1.060–3.347); *p* = 0.027). Patients with HCC with >50% liver involvement had a significantly shorter PFS than that of patients with <50% liver involvement (3.8 months (95% CI 1.6–6.6 months) vs. 7.5 months (95% CI 5.5–10.3 months), HR 2.288 (95% CI 1.231–4.252); *p* = 0.007). Patients who did not meet up to seven criteria had a significantly shorter PFS than that of patients did meet up to seven criteria (5.4 months (95% CI 3.5–6.9 months) vs. 10.5 months (95% CI 7.0–NE months), HR 2.847 (1.352–5.993); *p* = 0.004). Patients with portal vein invasion (Vp) had a significantly shorter PFS than that of patients without portal vein invasion (4.2 months (95% CI 1.8–5.4 months) vs. 7.6 months (95% CI; 5.8–9.0 months HR 2.223 (1.189–4.158); *p* = 0.009).

#### Treatment Response

We analyzed the response during treatment and at 6 weeks after atezolizumab and bevacizumab initiation by RECIST 1.1. and mRECIST (Table 2). As shown in Table 2, in the best response, 3 (2.9%), 17 (16.3%), 63 (60.6%), and 21 (20.2%) patients showed a complete response (CR), partial response (PR), stable disease (SD), and progressive disease (PD), respectively, based on RESICT v1.1. Thus, the objective response rate (ORR) and disease control rate (DCR) were 19.2% (20/104) and 79.8% (83/104), respectively. Similarly, in the mRECIST evaluation, 9 (8.7%), 20 (19.2%), 51 (49.0%), and 16 (15.4%) patients showed a CR, PR, SD, and PD, respectively, and the data for 8 (7%) patients were not obtained. Thus, the ORR and DCR were 27.9% (29/104) and 79.8% (83/104), respectively. The treatment responses at 6 weeks after atezolizumab and bevacizumab initiation are also shown in Table 2.

Subsequently, we compared the ORR and DCR between patients who did or did not meet the eligibility criteria of IMbrave150. As shown in Table 3, the ORR and DCR were similar between patients who did or did not meet the eligibility criteria of IMbrave150 in RECIST (*p* = 0.804, 18.4% vs. 19.7% and *p* = 1.000, 81.6% vs. 78.8%). Regarding the main factors causing patients to not meet the eligibility criteria of IMbrave150, ORR and DCR were similar between patients with or without history of systemic therapy, patients with or without platelet < 7.5 × 10^4^/μL, patients with or without 2+ proteinuria or more than 2+ proteinuria, and patients with or without Child-Pugh B. A subgroup analysis revealed that in RECIST v1.1, ORR was significantly higher in patients with up to 7-IN than in patients with up to 7—Out (40.7% vs. 11.7%, *p* = 0.003) (Table 3).

### 3.3. Changes in ALBI Score, mALBI Grade, and the Rate of Treatment Interruption

Subsequently, we evaluated the ALBI score and mALBI grade at baseline, 3, 6, 9, and 12 weeks after atezolizumab and bevacizumab initiation. Data for sequential changes in the ALBI score were available for 82 patients. As shown in Figure 3, similar to previous reports [22], the ALBI score deteriorated significantly at 3 weeks after treatment initiation (baseline vs. 3 weeks; 2.43 ± 0.45 vs. −2.34 ± 0.46; *p* = 0.012) but was restored to baseline levels at 6 weeks (baseline vs. 6 weeks; −2.43 ± 0.45 vs. −2.39 ± 0.51; *p* = 0.316). Thereafter, the ALBI score deteriorated gradually but significantly (baseline vs. 12 weeks; −2.43 ± 0.45 vs. −2.29 ± 0.54; *p* = 0.008). Between baseline and 12 weeks after treatment initiation, 26.8% of patients (22/82) showed a worsening in mALBI grade (Figure 3).

A subgroup analysis revealed that in patients who did not meet the IMbrave150 criteria, ALBI scores worsened significantly at 12 weeks after treatment initiation (baseline vs. 12 weeks; 2.35 ± 0.43 vs. −2.18 ± 0.53; *p* = 0.007), while in patients who did meet the IMbrave150 criteria, ALBI scores were similar between baseline and 12 weeks after treatment initiation (baseline vs. 12 weeks; −2.56 ± 0.43 vs. −2.48 ± 0.50; *p* = 0.363) (Figure 4). In addition, baseline ALBI scores tended to be deteriorated in patients who did not meet the eligibility criteria of IMbrave150 compared to patients who did meet the eligibility criteria of IMbrave150 (−2.35 ± 0.43 vs −2.56 ± 0.43 *p* = 0.051).

Among patients not meeting the eligibility criteria of IMbrave150, the ALBI score was significantly worse in patients with a history of systemic therapy than without (baseline vs. 12 weeks; −2.36 ± 0.46 vs. −2.24 ± 0.48, *p* = 0.037), while significant changes were not observed in patients who met the eligibility criteria of IMbrave150 (Figure 4). Appendix A summarizes the changes in ALBI scores in subgroups of patients with Platelet < 7.5 × 10^4^/μL, 2+ proteinuria or more than 2+ proteinuria, and Child-Pugh B.

Subsequently, we analyzed the rate of treatment interruption in patients who did or did not meet the IMbrave150 inclusion criteria. As shown in Table 4, the rate of atezolizumab interruption was significantly higher in patients who did not meet the inclusion criteria than in patients who met the inclusion criteria (22.2% (16/72) vs. 2.3% (1/46), *p* = 0.003). In addition, in patients with Child-Pugh grade B, the rate of atezolizumab interruption was significantly higher than that in patients with Child-Pugh A (12.3% (13/106) vs. 44.4% (4/9), *p* = 0.026). The precise reasoning behind treatment interruption in patients who did or did not meet the IMbrave150 eligibility criteria is demonstrated in Appendix A.

## 4. Discussion

In this real-world multicenter study, of 115 patients with unresectable HCC who were treated with atezolizumab and bevacizumab, 62.6% (72/115) did not meet the eligibility criteria of IMbrave150, largely due to a history of TKI treatment (83.3%, 60/72), platelet counts < 75 × 10^9^/L (9.7%, 7/72), Child-Pugh B (12.5%, 9/72); and 2+ proteinuria (11.1%, 8/72). We revealed that atezolizumab and bevacizumab were equally effective for patients who did or did not meet the eligibility criteria of IMbrave150 (median PFS 6.5 months (95% CI; 3.7–NE months) vs. 6.9 months (95% CI; 4.2–8.9 months), respectively, *p* = 0.794, with no difference in ORR and DCR between groups). Thus, even in patients who did not meet the eligibility criteria of IMbrave150, atezolizumab and bevacizumab treatment was highly effective.

Baseline ALBI scores tended to be worse in patients who did not meet the eligibility criteria of IMbrave150 than in patients who did meet the eligibility criteria of IMbrave150 (−2.35 ± 0.43 vs. −2.56 ± 0.43 *p* = 0.051). Furthermore, ALBI scores decreased significantly from baseline to 12 weeks after treatment initiation (2.35 ± 0.43 vs. −2.18 ± 0.54, respectively, *p* = 0.007) in patients who did not meet the IMbrave150 criteria, with no difference over time in patients who did meet the criteria. Maintaining the liver functional reserve is essential for subsequent salvage systematic therapy; accordingly, it may be necessary to carefully monitor changes in liver functional reserve during atezolizumab and bevacizumab therapy in patients who did not meet the IMbrave150 criteria.

A recent real-world study has reported that the median PFS is significantly shorter in patients who did not meet the IMbrave150 eligibility criteria than in patients who met the criteria [PFS: 3.7 months vs. 8.7 months] [11]. Although more than half of patients did not meet the eligibility criteria for IMbrave150 in this study, median PFS was similar between patients who did and those who did not meet the criteria (PFS 6.5 months (95% CI; 3.7–NE months) vs. 6.9 (95% CI; 4.2–8.9 months)). This discrepancy may be attributed to the difference in the number of patients who did not meet the IMbrave150 eligibility criteria, with deteriorated hepatic functional reserve. In the present study, there were 9 and 0 cases of Child-Pugh B and C, respectively, whereas the previous study included 35 (47.9%) Child-Pugh B cases and 6 (8.2%) Child-Pugh C cases not meeting the IMbrave150 eligibility criteria. Thus, the present study could reveal that, if hepatic functional reserve is preserved, atezolizumab and bevacizumab combination therapy is effective even for patients who did not meet the IMbrave150 inclusion criteria.

A recent study has revealed that OS and PFS are significantly shorter in patients with Child-Pugh B and C than in patients with Child-Pugh A [11,23]. In this study, the PFS and treatment response were similar between patients with Child-Pugh A and B. A small number of patients with Child-Pugh B could affect these results. In addition, most patients had a Child-Pugh score of 7. A recent study reports that the median PFS for atezolizumab and bevacizumab is similar in patients with a Child-Pugh score of 6 (5.1 months, 95%CI: 3.8–6.4 months) and 7 (6.3 months, 95% CI: 2.3–7.0 months) [24]. This might explain the better PFS of patients with Child-Pugh B in this study; however, further studies with larger sample sizes are required to confirm this.

In this study, as shown in Figure 2, the median PFS was similar between patients treated with atezolizumab and bevacizumab as a first-line therapy and as a second-line or later therapy. However, in patients who were treated with atezolizumab and bevacizumab as third line or later, the median PFS was significantly shorter than that in patients treated with atezolizumab and bevacizumab as first- or second-line systemic therapy. These results indicate that atezolizumab and bevacizumab should be selected as a first-line therapy or a second-line systemic therapy for unresectable HCC. It is not clear why patients treated with atezolizumab and bevacizumab as third-line or later line systematic therapy showed a shorter median PFS. Almost all drugs used as first- and second-line systematic therapy have anti-VEGF activity; thus, a lack of response to these drugs might be related to resistance to anti-VEGF therapy. A recent report has shown that anti-VEGF therapy could induce CD8+ T cell infiltration in HCC; therefore, immune checkpoint inhibitors combined with bevacizumab (e.g., atezolizumab and bevacizumab) could achieve a good response [25,26,27]. Additionally, interruption of bevacizumab tended to be higher in patients treated with atezolizumab and bevacizumab as third-line or later line systematic therapy, compared with those treated as first or second line (*p* = 0.09). Thus, HCC with resistance to anti-VEGF therapy might show a poor response to atezolizumab and bevacizumab. Further analyses are required to validate this hypothesis.

In this study, eight patients had 2+ or more than 2+ proteinuria, and most of these patients (7/8) had a history of TKI therapy. Although the sample size was limited, the median PFS, treatment response, and rate of treatment interruption in patients with 2+ or more than 2+ proteinuria were similar to those in patients without it. Thus, atezolizumab and bevacizumab might be effective and safe even in patients with 2+ or more than 2+ proteinuria.

There are several limitations to this retrospective multicenter study. The number of patients, especially patients with Child-Pugh B, 2+ or more than 2+ proteinuria and platelet < 7.5 × 10^4^/μL was limited, and this should be considered when interpreting the results. In addition, the observational period was relatively limited. Thus, a larger prospective study with a longer observational period is required in the near future to validate the results of this study.

## 5. Conclusions

In conclusion, atezolizumab and bevacizumab is effective even for patients who do not meet the IMbrave150 inclusion criteria. However, these patients showed a worse baseline liver functional reserve and decreases in median ALBI scores during atezolizumab and bevacizumab therapy, emphasizing the need for careful monitoring.

## Figures and Tables

**Figure 1 cancers-14-03938-f001:**
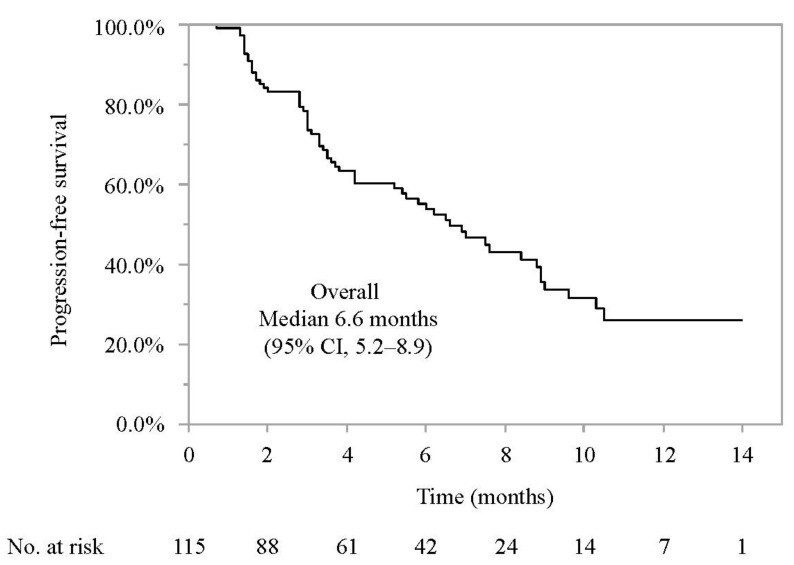
Progression-free survival in patients with unresectable HCC who were treated with atezolizumab and bevacizumab. Median progression-free survival was 6.6 months (95% confidence interval 5.2–8.9 months). 95% CI: 95% confidence interval.

**Figure 2 cancers-14-03938-f002:**
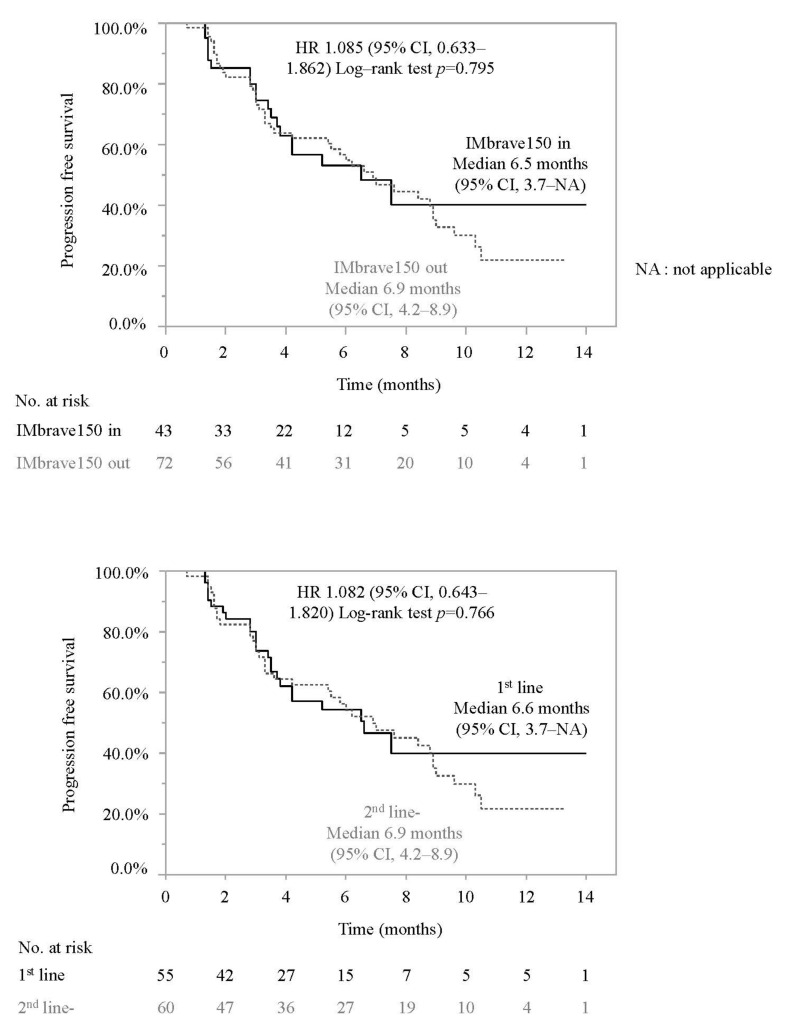
Comparison of progression-free survival in each subgroup. Survival curves for PFS were calculated by the Kaplan-Meier method and compared using the log-rank test. HR: Hazard Ratio, 95% CI: 95% confidence interval, NA: not applicable, BCLC: Barcelona Clinic Liver Cancer, mALBI: modified albumin-bilirubin grade, Vp: portal vein invasion.

**Figure 3 cancers-14-03938-f003:**
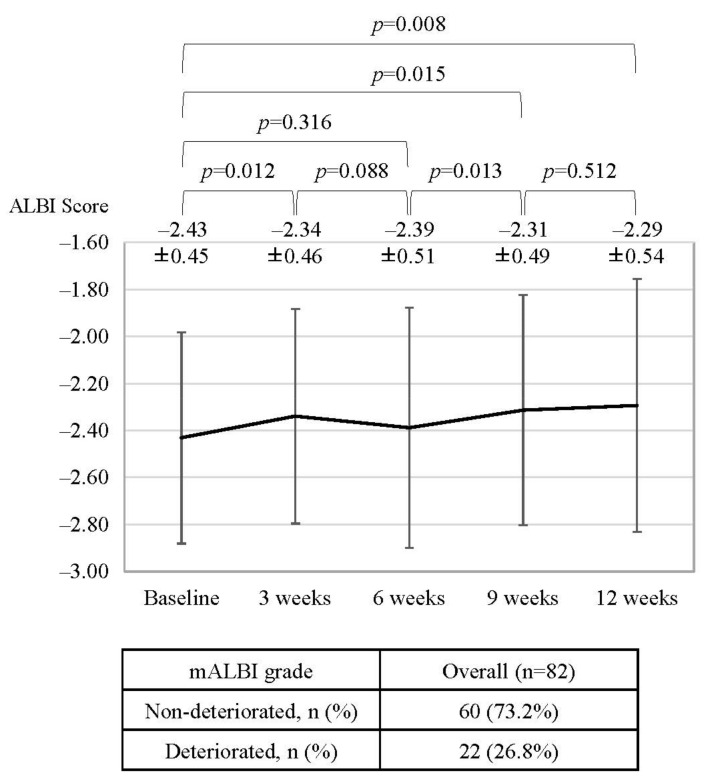
Changes in the ALBI score and mALBI grade during atezolizumab and bevacizumab treatment for unresectable HCC in all cohorts. mALBI: modified albumin-bilirubin grade.

**Figure 4 cancers-14-03938-f004:**
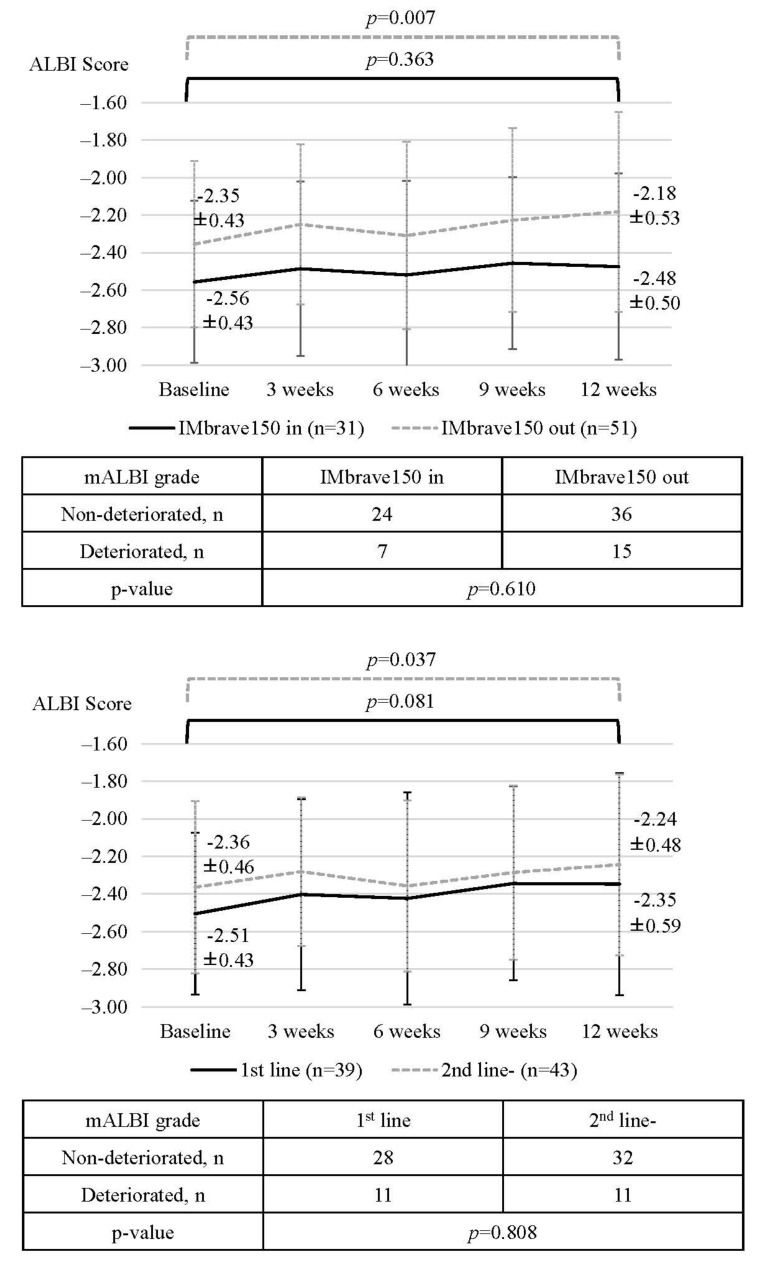
Comparison of changes in the ALBI score and mALBI grade during atezolizumab and bevacizumab treatment for unresectable HCC in subgroups stratified by eligibility criteria of IMbrave150 and history of systemic therapy. mALBI: modified albumin-bilirubin grade.

**Table 1 cancers-14-03938-t001:** Baseline patient characteristics.

Clinical Characteristics	Overall Cohort (n = 115)	Met the IMbrave150 Criteria (n = 43)	Did Not Meet the IMbrave150 Criteria (n = 72)	*p*-Value
Age, years (range)	72 (31–89)	73 (31–84)	72 (37–89)	0.744
Sex				
Male/Female	95 (82.6%)/20 (17.4%)	38 (88.4%)/5 (11.6%)	57 (79.2%)/15 (20.8%)	0.309
Etiology				
HBV	35 (30.4%)	11 (25.6%)	24 (33.3%)	0.411
HCV	21 (18.3%)	10 (23.3%)	11 (15.3%)	0.324
Non-viral	59 (51.3%)	22 (51.2%)	37 (51.4%)	1.000
ECOG PS				
0/1–2	92 (80.0%)/23 (20.0%)	39 (90.7%)/4 (9.3%)	53 (73.6%)/19 (26.4%)	0.031
BMI, kg/m^2^	23.6 (15.9–37.7)	24.0 (18.7–37.7)	23.3 (15.9–33.0)	0.152
Proteinuria				
0–1+/2+	98 (85.2%)/8 (7.0%)	41 (89.1%)/0 (0.0%)	57 (79.2%)/8 (11.1%)	0.022
White blood cell, mm^3^	4920 (1970–12780)	5300 (2950–12,780)	4900 (1970–11,800)	0.043
Neutrophil count, mm^3^	3203 (1185–9971)	3380 (1503–9204)	2978 (1185–9971)	0.098
Lymphocyte count, mm^3^	1168 (140–2881)	1207 (559–2657)	1100 (140–2881)	0.056
Neutrophil/Lymphocyte ratio	2.70 (0.83–18.68)	3.04 (0.83–7.92)	2.57 (0.98–18.68)	0.967
Platelet, ×10^9^/L	162 (36–586)	154 (77–558)	167 (36–586)	0.773
Prothrombin time, %	92.9 (35.3–150.0)	90.0 (42.6–116.9)	94.6 (35.3–150.0)	0.317
NH3, µg/dL	42 (8–136)	42 (17–116)	42 (8–136)	0.538
Albumin, g/dL	3.7 (2.6–4.8)	3.8 (2.9–4.8)	3.7 (2.6–4.8)	0.030
Total bilirubin, mg/dL	0.8 (0.2–3.8)	0.8 (0.2–2.9)	0.8 (0.3–3.8)	0.979
mALBI grade				
½	38 (33.0%)/77 (67.0%)	17 (39.5%)/26 (60.5%)	21 (29.2%)/51 (70.8%)	0.307
1	38 (33.0%)	17 (39.5%)	21 (29.2%)	0.307
2a	37 (32.2%)	15 (34.8%)	22 (30.6%)	0.683
2b	40 (34.8%)	11 (25.6%)	29 (40.3%)	0.156
AST, IU/L	42 (14–672)	35 (16–128)	47 (14–672)	0.164
ALT, IU/L	28 (7–278)	25 (8–122)	33 (7–289)	0.256
Child-Pugh Grade				
A/B	106 (92.2%)/9 (7.8%)	43 (100.0%)/0 (0.0%)	63 (87.5%)/9 (12.5%)	0.025
Child-Pugh Score				
5	62 (53.9%)	24 (55.8%)	38 (52.8%)	0.847
6	44 (38.3%)	19 (44.9%)	25 (34.7%)	0.329
7	6 (5.2%)	0 (0.0%)	6 (8.3%)	0.082
8	3 (2.6%)	0 (0.0%)	3 (4.2%)	0.292
AFP, ng/mL *	74.2 (0.8–1,450,000.0)	51.6 (0.8–591,315.4)	77.2 (1.1–14,500,000.0)	0.817
AFP > 400	40 (34.8%)	15 (34.9%)	25 (34.7%)	1.000
DCP, mAU/mL *	924 (11–245,000)	509 (21–213,066)	1787 (11–245,000)	0.035
Maximum intrahepatic tumor size, mm	36 (0–220)	30 (0–167)	38 (0–220)	0.307
More than 50% liver involvement	16 (17.7%)	4 (9.3%)	12 (16.7%)	0.405
Diffuse type	15 (15.6%)	5 (11.6%)	10 (13.9%)	0.784
Number of intrahepatic tumors				
None	14 (12.2%)	5 (11.6%)	9 (12.5%)	1.000
1	11 (9.6%)	5 (11.6%)	6 (8.3%)	0.745
Multiple	90 (78.3%)	33 (76.7%)	57 (79.2%)	0.817
BCLC stage				
B/C	35 (30.4%)/80 (69.6%)	13 (30.2%)/30 (69.8%)	22 (30.6%)/50 (69.4%)	1.000
Up-to-7 in/out	30 (26.1%)/85 (73.9%)	10 (23.3%)/33 (76.7%)	20 (27.8%)/52 (72.2%)	0.665
Positive for Vp	23 (20.0%)	11 (25.6%)	12 (16.7%)	0.335
Vp4	4 (3.5%)	2 (4.7%)	2 (2.8%)	0.629
Positive for Vv	5 (4.3%)	1 (2.3%)	4 (5.6%)	0.649
Positive for bile duct invasion	4 (3.5%)	1 (2.3%)	3 (4.2%)	1.000
Positive for LN metastasis	20 (17.4%)	7 (16.8%)	13 (18.1%)	1.000
Positive for EHM	46 (40.0%)	17 (39.5%)	29 (40.3%)	1.000
History of varices treatment	8 (7.0%)	1 (2.3%)	7 (9.7%)	0.255
History of hypertension	69 (60.0%)	27 (62.8%)	42 (58.3%)	0.697
Naïve/recurrence	14 (20.9%)/91 (79.1%)	15 (34.9%)/28 (65.1%)	9 (12.5%)/63 (87.5%)	0.008
History of operation	59 (51.3%)	21 (48.8%)	38 (52.8%)	0.707
History of RFA	41 (35.7%)	14 (32.6%)	25 (34.7%)	0.842
History of TACE	58 (50.4%)	14 (32.6%)	44 (61.1%)	0.004
1st line systemic chemotherapy	55 (47.8%)	43 (100.0%)	12 (16.7%)	<0.001
2nd line	41 (35.7%)	0 (0.0%)	41 (56.9%)	
3rd line	19 (16.5%)	0 (0.0%)	19 (26.4%)	
History of TKI	60 (52.2%)	0 (0.0%)	60 (83.3%)	
Sorafenib	19 (16.5%)	0 (0.0%)	19 (27.5%)	
Regorafenib	8 (7.0%)	0 (0.0%)	8 (11.1%)	
Lenvatinib	59 (51.3%)	0 (0.0%)	59 (81.9%)	
Observation period, months *	6.8 (0.1–15.4)	5.6 (0.1–14.9)	7.2 (0.3–15.4)	0.140

* Data are presented as median (range) or n. Abbreviations: HCV: hepatitis C virus, HBV: hepatitis B virus, ECOG PS: Eastern Cooperative Oncology Group performance status, BMI: body mass index, AST: aspartate transaminase, ALT: alanine aminotransferase, mALBI grade: modified albumin–bilirubin grade, AFP: alpha-fetoprotein, EHM: extrahepatic metastasis, TKI: tyrosine kinase inhibitor, DCP: des-gamma-carboxy prothrombin, BCLC: The Barcelona Clinic Liver Cancer, Vp: portal vein invasion, Vv: hepatic vein invasion, LN: lymph node, RFA: Radiofrequency ablation, TACE: Transcatheter arterial chemoenbolzation.

**Table 2 cancers-14-03938-t002:** Comparison of progression-free survival in each subgroup.

	RECIST v1.1		mRECIST	
	6 weeks	Best response	6 weeks	Best response
CR, n (%)	0 (0.0)	3 (2.9)	2 (1.9)	9 (8.7)
PR, n (%)	9 (8.7)	17 (16.3)	18 (17.3)	20 (19.2)
SD, n (%)	74 (71.2)	63 (60.6)	61 (58.7)	51 (49.0)
PD, n (%)	18 (17.3)	21 (20.2)	12 (12.5)	16 (15.4)
NE, n (%)	3 (2.9)	0 (0.0)	10 (9.6)	8 (7.7)
ORR, n (%)	9 (8.7)	20 (19.2)	20 (19.2)	29 (27.9)
DCR, n (%)	83 (79.8)	83 (79.8)	81 (77.9)	80 (76.9)

Abbreviations: CR: complete response, PR: partial response, SD: stable disease, PD: progressive disease, NE: not evaluable, ORR: objective response rate, DCR: disease control rate.

**Table 3 cancers-14-03938-t003:** Comparison of clinical responses in patients who were treated with atezolizumab and bevacizumab in each subgroup.

	RECIST v1.1			mRECIST		
	IMbrave150 in (n = 38)	IMbrave150 out (n = 66)	*p*-Value	IMbrave150 in (n = 38)	IMbrave150 out (n = 66)	*p*-Value
ORR (%)	18.4%	19.7%	1.000	29.0%	27.3%	1.000
DCR (%)	81.6%	78.8%	0.804	73.7%	78.8%	0.631
	1st line (n = 48)	2nd line- (n = 56)	*p*-value	1st line (n = 48)	2nd line- (n = 56)	*p*-value
ORR (%)	18.8%	19.6%	1.000	31.3%	25.0%	0.517
DCR (%)	81.3%	78.6%	0.809	75.0%	78.6%	0.816
	Child-Pugh A (n = 96)	Child-Pugh B (n = 8)	*p*-value	Child-Pugh A (n = 96)	Child-Pugh B (n = 8)	*p*-value
ORR (%)	20.8%	0.0%	0.349	28.1%	25.0%	1.000
DCR (%)	79.2%	87.5%	1.000	77.1%	75.0%	1.000
	Proteinuria 0-1+ (n = 91)	Proteinuria 2+ (n = 7)	*p*-value	Proteinuria 0-1+ (n = 91)	Proteinuria 2+ (n = 7)	*p*-value
ORR (%)	18.7%	28.6%	0.618	27.5%	28.6%	1.000
DCR (%)	82.4%	85.7%	1.000	79.1%	85.7%	1.000
	Platelet ≧ 7.5 × 10^4^/μL (n = 98)	Platelet < 7.5 × 10^4^/μL (n = 6)	*p*-value	Platelet ≧ 7.5 × 10^4^/μL (n = 98)	Platelet < 7.5 × 10^4^/μL (n = 6)	*p*-value
ORR (%)	18.4%	33.3%	0.325	26.5%	50.0%	0.345
DCR (%)	79.6%	83.3%	1.000	77.6%	66.7%	0.620
	1st–2nd line (n = 85)	3rd line (n = 19)	*p*-value	1st–2nd line (n = 85)	3rd line (n = 19)	*p*-value
ORR (%)	21.2%	10.5%	0.355	31.8%	10.5%	0.089
DCR (%)	78.8%	84.2%	0.758	74.1%	89.5%	0.230
	Viral (n = 51)	Non-viral (n = 53)	*p*-value	Viral (n = 51)	Non-viral (n = 53)	*p*-value
ORR (%)	17.7%	20.8%	0.805	23.5%	32.1%	0.386
DCR (%)	80.4%	79.3%	1.000	72.6%	81.1%	0.356
	BCLC-B (n = 41)	BCLC-C (n = 63)	*p*-value	BCLC-B (n = 41)	BCLC-C (n = 63)	*p*-value
ORR (%)	19.5%	19.0%	1.000	34.2%	23.8%	0.271
DCR (%)	87.5%	76.4%	0.290	87.5%	72.2%	0.130
	mALBI 1-2a (n = 73)	mALBI 2b (n = 31)	*p*-value	mALBI 1-2a (n = 73)	mALBI 2b (n = 31)	*p*-value
ORR (%)	21.9%	12.9%	0.416	31.5%	19.4%	0.240
DCR (%)	78.1%	83.9%	0.600	76.7%	77.4%	1.000
	<50% liver involvement (n = 90)	≧50% liver involvement (n = 14)	*p*-value	<50% liver involvement (n = 90)	≧50% liver involvement (n = 14)	*p*-value
ORR (%)	20.0%	14.3%	1.000	26.0%	14.3%	0.340
DCR (%)	80.0%	78.6%	1.000	76.7%	78.6%	1.000
	Up-to-7 in (n = 27)	Up-to-7 out (n = 77)	*p*-value	Up-to-7 in (n = 27)	Up-to-7 out (n = 77)	*p*-value
ORR (%)	40.7%	11.7%	0.003	48.2%	20.8%	0.002
DCR (%)	88.9%	76.6%	0.265	85.2%	74.0%	0.296
	Vp − (n = 88)	Vp + (n = 16)	*p*-value	Vp − (n = 88)	Vp + (n = 16)	*p*-value
ORR (%)	21.6%	6.3%	0.298	31.8%	6.3%	0.037
DCR (%)	83.0%	62.5%	0.087	79.6%	62.5%	0.194

Abbreviations: ORR: objective response rate, DCR: disease control rate, BCLC: Barcelona Clinic Liver Cancer, mALBI: modified albumin-bilirubin grade, Vp: portal vein invasion.

**Table 4 cancers-14-03938-t004:** Comparison of the rate of treatment discontinuation and interruption in patients who were treated with atezolizumab and bevacizumab in each subgroup.

	Overall		
Discontinuation due to AE n, (%)	7 (6.1)		
Interruption of Atezo n, (%)	17 (14.8)		
Interruption of Bev n, (%)	30 (26.1)		
	IMbrave150 in (n = 43)	IMbrave150 out (n = 72)	*p*-value
Discontinuation due to AE n, (%)	1 (2.3)	6 (8.3)	0.254
Interruption of Atezo n, (%)	1 (2.3)	16 (22.2)	0.003
Interruption of Bev n, (%)	8 (18.6)	22 (30.6)	0.191
	1st line (n = 55)	2nd line (n = 60)	*p*-value
Discontinuation due to AE n, (%)	4 (7.3)	3 (5.0)	0.710
Interruption of Atezo n, (%)	5 (9.1)	12 (20.0)	0.120
Interruption of Bev n, (%)	13 (23.6)	17 (28.3)	0.672
	Child-Pugh A (n = 106)	Child-Pugh B (n = 9)	*p*-value
Discontinuation due to AE n, (%)	6 (5.7)	1 (11.1)	0.444
Interruption of Atezo n, (%)	13 (12.3)	4 (44.4)	0.026
Interruption of Bev n, (%)	26 (24.5)	4 (44.4)	0.237
	Proteinuria 0-1+ (n = 98)	Proteinuria 2+ (n = 8)	*p*-value
Discontinuation due to AE n, (%)	6 (6.1)	1 (12.5)	0.423
Interruption of Atezo n, (%)	14 (14.3)	2 (25.0)	0.347
Interruption of Bev n, (%)	26 (26.5)	3 (37.5)	0.681
	Platelet ≧ 7.5 × 10^4^/μL (n =108)	Platelet < 7.5 × 10^4^/μL (n = 7)	*p*-value
Discontinuation due to AE n, (%)	7 (6.5)	0 (0.0)	1.000
Interruption of Atezo n, (%)	14 (13.0)	3 (42.9)	0.065
Interruption of Bev n, (%)	26 (24.1)	4 (57.1)	0.075
	1st–2nd line (n = 85)	3rd line (n = 19)	*p*-value
Discontinuation due to AE n, (%)	7 (7.3)	0 (0.0)	0.598
Interruption of Atezo n, (%)	13 (13.5)	4 (21.1)	0.478
Interruption of Bev n, (%)	22 (22.9)	8 (42.1)	0.093

Abbreviations: AE: adverse event, Atezo: Atezolizumab, Bev: bevacizumab, BCLC: Barcelona Clinic Liver Cancer, mALBI: modified albumin-bilirubin grade, Vp: portal vein invasion.

## Data Availability

The data that support the findings of this study are available from the corresponding author upon reasonable request.

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
