# Peer review of "Efficacy and Effect on Liver Functional Reserve of Atezolizumab and Bevacizumab for Unresectable Hepatocellular Carcinoma in Patients Who Do Not Meet Eligibility Criteria of IMbrave150"

_cancers, 2022, doi:10.3390/cancers14163938_

Round 1

Reviewer 1 Report

In this manuscript, Sho et al. have conducted a real-word multi-center study to evaluate the clinical outcome of HCC patients treated with atezo-bev beyond the eligibility criteria of IM brave 150 study. The study concluded that atezo-bev has similar efficacy in patients beyond the eligibility criteria but associated with higher risks of liver function deterioration. 

Study design: This is a retrospective study that prone to some intrinsic biases that have been mentioned by the authors. 

Study novelty and interest to general audience: Fair novelty, but important data to practicing clinicians as its uncommon to take care patients who do not meet the IMbrave 150 eligibility criteria

Are the claims convincing? Are the claims fully supported by the experimental data? 

1) Authors concluded that IMbrave-out was associated with similar efficacy as IMbrave-in; however, the patients with Child-Pugh B, Proteinuria 2+, platelet < 75 was indeed under-represented. It was rather difficult to draw any meaningful conclusion based on < 10 patients in each of these subgroups. I agree that further authors’ suggestions that further large-scale study is needed to confirm this claim. I would think the present findings only can support the claim that atezo-bev is equally effective in 1st line vs. 2nd line. 

2) Treatment interruption was significantly more common among patients of IMbrave 150-out, in particular among patients with Child-Pugh B and low platelet. Could authors provide more information regarding to the reasons of treatment interruption? Was it because of the variceal bleeding or other reasons? 

3) Patients were required to have EGD assessment within 6 months of study enrollment to screen for any varices. However, author didn’t mention if study patients have EGD assessment or not in current study. 

Statistical design: Appropriate 

Figures and tables: Well presented in good quality 

Ethical concerns: None

Discussion: The author should put more emphasis on compare the discrepancy of current study findings with another recent one (Ther Adv Med Oncol 2022, 14, 17588359221080298, 413 doi:10.1177/17588359221080298) 

Author Response

August 08, 2022

Prof. Dr. Samuel C. Mok

Editor-in-Chief

Cancers

Dear prof. Mok,

We wish to re-submit the manuscript titled “Efficacy and effect on liver functional reserve of atezolizumab and bevacizumab for unresectable hepatocellular carcinoma in patients who do not meet eligibility criteria of IMbrave150.” The manuscript ID is: cancers-1829074.

We greatly appreciate the comments of the editor and reviewers; addressing them has significantly improved the quality of the manuscript. We hope that the revised manuscript is now suitable for publication in Cancers.

 The manuscript has been rechecked and the necessary changes have been made in accordance with the reviewers’ suggestions. Our point-by-point responses to the editor and reviewers’ comments are attached. Changes to the text are highlighted in red font for the reviewers’ convenience.

Thank you for your consideration. I look forward to hearing from you.

Sincerely,

Goki Suda, M.D., Ph.D

Department of Gastroenterology and Hepatology/Graduate School of Medicine

Hokkaido University, North 15, West 7, Kita-ku, Sapporo, Hokkaido 060-8638, Japan

Phone number: +81 11-716-1161

Fax number: +81 11-706-7867

Email address: [email protected]

Our point-by-point responses to the reviewers’ comments and suggestions are listed below:

Responses to Reviewers:

Reviewer 1

In this manuscript, Sho et al. have conducted a real-word multi-center study to evaluate the clinical outcome of HCC patients treated with atezo-bev beyond the eligibility criteria of IM brave 150 study. The study concluded that atezo-bev has similar efficacy in patients beyond the eligibility criteria but associated with higher risks of liver function deterioration. 

Study design: This is a retrospective study that prone to some intrinsic biases that have been mentioned by the authors. 

Study novelty and interest to general audience: Fair novelty, but important data to practicing clinicians as its uncommon to take care patients who do not meet the IMbrave 150 eligibility criteria

[Response]

We thank you for your constructive and insightful comments and suggestions. The reviewer’s suggestion has significantly improved the quality of the manuscript. We hope that the revised manuscript is now suitable for publication in Cancers.

We addressed each of the mentioned issues in details below.

1) Authors concluded that IMbrave-out was associated with similar efficacy as IMbrave-in; however, the patients with Child-Pugh B, Proteinuria 2+, platelet < 75 was indeed under-represented. It was rather difficult to draw any meaningful conclusion based on < 10 patients in each of these subgroups. I agree that further authors’ suggestions that further large-scale study is needed to confirm this claim. I would think the present findings only can support the claim that atezo-bev is equally effective in 1st line vs. 2nd line. 

[Response]

We appreciate your constructive and helpful comments and suggestions. As pointed out by the reviewer, we agree that the number of patients with Child-Pugh B, Proteinuria 2+, or platelet count < 75 × 109/L was quite limited.

Therefore, we have specified this as an important study limitation and mentioned that this should be considered when interpreting the results. (Refer to lines 354 to 356)

2) Treatment interruption was significantly more common among patients of IMbrave 150-out, in particular among patients with Child-Pugh B and low platelet. Could authors provide more information regarding to the reasons of treatment interruption? Was it because of the variceal bleeding or other reasons? 

Response

We thank you for your constructive and insightful comments and suggestions. As per the reviewer’s suggestion, we have added the precise reasons pertaining to treatment interruption observed in patients with or without Child-pugh B and low platelet count. We added these in the Supplementary Table S1 and further described them in lines 282 to 284.

3) Patients were required to have c of study enrollment to screen for any varices. However, author didn’t mention if study patients have EGD assessment or not in current study. 

Response

We appreciate the reviewer’s constructive comments and suggestions. As pointed out by the reviewer, we have added the description that “All included patients were evaluated, using endoscopy, for the presence of varices before initiation of atezolizumab and bevacizumab, and, when necessary, the varices were properly treated.” in lines 96 to 98.

Statistical design: Appropriate 

Figures and tables: Well presented in good quality 

Ethical concerns: None

Discussion: The author should put more emphasis on compare the discrepancy of current study findings with another recent one (Ther Adv Med Oncol 2022, 14, 17588359221080298, 413 doi:10.1177/17588359221080298) 

Response

We thank you for your insightful suggestions. As pointed out, we have put more emphasis on comparing the discrepancy of the current study findings with the aforementioned study (Ther Adv Med Oncol 2022, 14, 17588359221080298, 413 doi:10.1177/17588359221080298) in lines 308 to 320 as follows:

A recent real-world study has reported that the median PFS is significantly shorter in patients who did not meet the IMbrave150 eligibility criteria than in patients who met the criteria [PFS: 3.7 months vs. 8.7 months]. Although more than half of patients did not meet the eligibility criteria for IMbrave150 in this study, median PFS was similar between patients who did and those who did not meet the criteria (PFS 6.5 months (95% CI; 3.7–NE months) vs. 6.9 (95% CI; 4.2–8.9 months)). This discrepancy might be attributed to the difference in the number of patients, who did not meet the IMbrave150 eligibility criteria, with deteriorated hepatic functional reserve. In the present study, there were 9 and 0 cases of Child-Pugh B and C, respectively, whereas the previous study included 35 (47.9%) Child-Pugh B cases and 6 (8.2%) Child-Pugh C cases not meeting the IMbrave150 eligibility criteria. Thus, the present study might reveal that, if hepatic functional reserve is preserved, atezolizumab and bevacizumab combination therapy is effective even for patients who did not meet the IMbrave150 inclusion criteria.

Reviewer 2 Report

We read with interest the paper entitled “Efficacy and effect on liver functional reserve of atezolizumab 2 and bevacizumab for unresectable hepatocellular carcinoma in 3 patients who do not meet eligibility criteria of IMbrave150” by Takuya Sho et al.

It is a retrospective study carried out in the real world setting of unresectable HCC comparing two groups of patient meeting or non-meeting IMbrave150 study eligibility criteria and treated with atezolizumab  and bevacizumab. Data of the study demonstrated that combo therapy showed similar efficacy in both groups.

Although the study is retrospective , the groups were well balanced as to the most important clinic and laboratory characteristics making their comparison feasible and correct. Interestingly, as the authors stated, the study was carried out to depict the efficacy of this treatment in an “in field” scenario which may be somehow different from the scenario depicted by a RCT. However, data on PFS are very similar to those obtained in the IMbrave 150 original study. In addition, this study provided relevant information on the response of combo therapy when used as third or further line of therapy, suggesting that atezolizumab  and bevacizumab should be used as soon as possible as first line of therapy in order not to lose its efficacy. Furthermore, the study rises  a warning  on the possible deterioration of liver function, as depicted by the ALBI score monitoring, in patients with more advanced liver disease causing treatment discontinuation.

One point needs to be addressed. The authors did not take in consideration OS. This should be explained in the statistical section of Mat&Met.

Author Response

August 08, 2022

Prof. Dr. Samuel C. Mok

Editor-in-Chief

Cancers

Dear prof. Mok,

We wish to re-submit the manuscript titled “Efficacy and effect on liver functional reserve of atezolizumab and bevacizumab for unresectable hepatocellular carcinoma in patients who do not meet eligibility criteria of IMbrave150.” The manuscript ID is: cancers-1829074.

We greatly appreciate the comments of the editor and reviewers; addressing them has significantly improved the quality of the manuscript. We hope that the revised manuscript is now suitable for publication in Cancers.

 The manuscript has been rechecked and the necessary changes have been made in accordance with the reviewers’ suggestions. Our point-by-point responses to the editor and reviewers’ comments are attached. Changes to the text are highlighted in red font for the reviewers’ convenience.

Thank you for your consideration. I look forward to hearing from you.

Sincerely,

Goki Suda, M.D., Ph.D

Department of Gastroenterology and Hepatology/Graduate School of Medicine

Hokkaido University, North 15, West 7, Kita-ku, Sapporo, Hokkaido 060-8638, Japan

Phone number: +81 11-716-1161

Fax number: +81 11-706-7867

Email address: [email protected]

Our point-by-point responses to the reviewers’ comments and suggestions are listed below:

Responses to Reviewers:

Reviewer 2

We read with interest the paper entitled “Efficacy and effect on liver functional reserve of atezolizumab 2 and bevacizumab for unresectable hepatocellular carcinoma in 3 patients who do not meet eligibility criteria of IMbrave150” by Takuya Sho et al.

It is a retrospective study carried out in the real world setting of unresectable HCC comparing two groups of patient meeting or non-meeting IMbrave150 study eligibility criteria and treated with atezolizumab  and bevacizumab. Data of the study demonstrated that combo therapy showed similar efficacy in both groups.

Although the study is retrospective , the groups were well balanced as to the most important clinic and laboratory characteristics making their comparison feasible and correct. Interestingly, as the authors stated, the study was carried out to depict the efficacy of this treatment in an “in field” scenario which may be somehow different from the scenario depicted by a RCT. However, data on PFS are very similar to those obtained in the IMbrave 150 original study. In addition, this study provided relevant information on the response of combo therapy when used as third or further line of therapy, suggesting that atezolizumab and bevacizumab should be used as soon as possible as first line of therapy in order not to lose its efficacy. Furthermore, the study rises a warning on the possible deterioration of liver function, as depicted by the ALBI score monitoring, in patients with more advanced liver disease causing treatment discontinuation.

One point needs to be addressed. The authors did not take in consideration OS. This should be explained in the statistical section of Mat&Met.

Response

We appreciate your constructive and helpful comments and suggestions. As pointed out, we have added the description of OS in the Material and Methods section. (Refer to lines 131 and 132).

The reviewers’ suggestions have significantly helped in improving the overall quality of the manuscript. We hope that the revised manuscript is now suitable for publication in Cancers.
